

# Long-term effects of straw and straw-derived biochar on soil aggregation and fungal community in a rice–wheat rotation system

Naling Bai[1,2,*], Hanlin Zhang[1,2,*], Shuangxi Li[1,2], Xianqing Zheng[1,2], Juanqin Zhang[1,2], Haiyun Zhang[1,2], Sheng Zhou[1,2], Huifeng Sun[1,2] and Weiguang Lv[1,2,3]

[1] Eco-environmental Protection Research Institute, Shanghai Academy of Agricultural Science, Shanghai, China
[2] Agricultural Environment and Farmland Conservation Experiment Station of Ministry Agriculture, Shanghai, China
[3] Shanghai Key Laboratory of Protected Horticultural Technology, Shanghai, China
* These authors contributed equally to this work.

Corresponding author
Weiguang Lv,
lvweiguang@saas.sh.cn

## ABSTRACT

**Background:** Soil aggregation is fundamental for soil functioning and agricultural productivity. Aggregate formation depends on microbial activity influencing the production of exudates and hyphae, which in turn act as binding materials. Fungi are also important for improving soil quality and promoting plant growth in a symbiotic manner. There is a scarcity of findings comparing the long-term impacts of different yearly double-crop straw return modes (e.g., straw return to the field and straw-derived biochar return to the field) on soil aggregation and fungal community structure in rice–wheat rotation systems.

**Methods:** The effects of 6-year continuous straw and straw-derived biochar amendment on soil physicochemical properties and the fungal community were evaluated in an intensively managed crop rotation system (rice–wheat). Soil samples of different aggregates (macroaggregates, microaggregates, and silt clay) from four different fertilization regimes (control, CK; traditional inorganic fertilization, CF; straw returned to field, CS; straw-derived biochar addition, CB) were obtained, and Illumina MiSeq sequencing analysis of the fungal internal transcribed spacer gene was performed.

**Results:** Compared to CF, CS and CB enhanced soil organic carbon, total nitrogen, and aggregation in 0–20 and 20–40 cm soil, with CB exhibiting a stronger effect. Additionally, agrowaste addition increased the mean weight diameter and the geometric diameter and decreased the fractal dimension ($p < 0.05$). Principal coordinates analysis indicated that fertilization management affected fungal community structure and aggregation distribution. In addition, CS increased fungal community richness and diversity, compared to CK, CB decreased these aspects. Ascomycota, unclassified_k_Fungi, and Basidiomycota were the dominant phyla in all soil samples. At the genus level, CB clearly increased fungi decomposing biosolids (*Articulospora* in macroaggregates in 0–20 cm soil and *Neurospora* in macroaggregates in 20–40 cm soil); decreased pathogenic fungi (*Monographella* in macroaggregates and *Gibberella* in microaggregates in 0–20 cm soil) and

$CO_2$-emission-related fungi (*Pyrenochaetopsis* in microaggregates and silt clay in 0–40 cm soil) ($p < 0.05$). Straw and biochar with inorganic fertilizer counteracted some of the adverse effects of the inorganic fertilizer with biochar showing better effects than straw.

# INTRODUCTION

Soil aggregation, which is largely responsible for soil structure, is fundamental for soil functioning and agricultural productivity (*Zhao et al., 2018*). Aggregate formation depends on microbial activity influencing the production of exudates and hyphae, which in turn act as binding materials. Generally, soil aggregate dynamics are influenced by a number of factors, including (i) the soil biota (both microorganisms and macrofauna), (ii) the availability of inorganic binding agents, (iii) root growth, (iv) soil mineralogy and texture, and (v) environmental conditions (*Six et al., 2004*).

Soil microbial community abundance and structure have been widely used to indicate soil quality changes since they are sensitive to environmental changes. Fungi are also important soil microorganisms as, just like bacteria, they are capable of decomposing complex carbon compounds, improving soil quality, and promoting plant growth in a symbiotic manner. A fungus-dominant microbial community is important for carbon (C) stabilization and can produce more protected and stable C storage (*Six et al., 2006*). The microbial community is greatly affected by anthropogenic activities, such as agricultural intensification and fertilization (*Ding et al., 2017*). In long-term fertilizer experiments, the community structure, population and function of microorganisms are all affected (*Cinnadurai, Gopalaswamy & Balachandar, 2013*). For example, nitrogen (N) fertilizer and N plus Phosphate (P) fertilizers increase the size of fungi, reduce fungal biodiversity and change the community composition (*Zhou et al., 2016*). A combination of organic amendment and inorganic fertilizer not only recycles resources and improves soil fertility but also impacts soil fungal diversity (*Ding et al., 2017*).

Straw is rich in organic carbon, nitrogen, phosphorus, potassium, silicon, and other mineral nutrients. Crop straw can be used for cellulosic feedstock, renewable energy, paper-making, and animal feed. However, in most cases, straw has been usually either burned or discarded, which is a great waste of raw materials. Effective utilization of agrowaste, for example, straw, is a priority issue for the sustainable development of resources, environment, and agriculture (*Cui et al., 2017*). Relative to a control, straw return significantly increased the proportion of macroaggregates in the 0–20 cm soil layer and the soil organic carbon (SOC) content in each soil aggregate size class (*Zhao et al., 2018*). Straw and straw-derived biochar have attracted increasing attention in China; furthermore, the pyrolysis conversion of crop straw into biochar avoids the problem of the slow decomposition rate of straw in soil. Furthermore, biochar increases soil carbon sequestration more than straw (*Cui et al., 2017*).

The nutrients and effects of biochar can vary considerably depending on the feedstock, pyrolysis conditions, and application environment. Therefore, the interactions between biochar and microorganisms might be quite complicated. Generally, biochar holds promise as an amendment for soil quality improvement, sequestration of atmosphere $CO_2$, and increasing crop productivity (*Feng et al., 2012*). Some studies also revealed that biochar addition can negatively impact soil properties. For example, *Warnock et al. (2010)* reported that biochar (lodgepole pine, peanut shell, and mango wood) application decreased arbuscular mycorrhizal fungal abundance in roots or soils. A lower microbial metabolic quotient and enzyme activity were observed in a rice paddy 4 years after a single incorporation of 20 and 40 t ha$^{-1}$ biochar (*Zheng et al., 2016*). Additionally, *Jindo et al. (2012)* found that biochar addition increased the microbial community and influenced the performance of the composting process. Fungal community composition was more affected by biochar than bacterial community composition (*Zheng et al., 2016*). *Abujabhah et al. (2016)* noted that in orchard soil, the effect of biochar and compost addition on fungal community structure alteration was the most highly significant, whereas a smaller difference was observed for the overall bacterial community structure.

Numerous studies have shown the short-term impacts of either straw or biochar on the soil microbial community (*Ameloot et al., 2013*; *Wang et al., 2017a*). In finer-textured Yolo soil, 1% walnut shell biochar decreased the fungi to bacteria ratio with an increase in total phosphlipid fatty acids (PLFA) (*Wang et al., 2017a*). *Chen et al. (2017)* found that under short-term straw return, straw application significantly increased total PLFA, bacterial biomass, and actinomycete biomass in Jiangyan, but had no significant effects on PLFAs in both Qujialing and Guangde, relative to no straw return treatment. However, to our knowledge, there is a scarcity of findings comparing the long-term impacts of different yearly double-crop straw return modes (e.g., straw return to the field and straw-derived biochar return to the field) on soil aggregate and fungal community structure in rice–wheat rotation systems. Thus, our objectives were to apply Illumina MiSeq high-throughput sequencing to analyze (i) the influence of straw and straw-derived biochar on aggregate distribution and stabilization and (ii) the effects of fertilization management and aggregation on fungal community structure.

## MATERIALS AND METHODS

### Experimental design

An in situ 6-year field study was operated with a typical rice–wheat rotation system in a sandy loam soil at Zhuanghang Field Station, Shanghai Academy of Agricultural Sciences, China (30°53′N, 121°23′E). Treatments were denoted in triplicate as follows: (i) CK (no fertilizer application), (ii) CF (conventional inorganic fertilizer application), (iii) CS (the same total pure nitrogen application as in CF plus three t ha$^{-1}$ straw returning to the field), and (iv) CB (the same total pure nitrogen application as in CF plus one t ha$^{-1}$ straw-derived biochar returning to the field). The experimental plots were arranged in a completely randomized block with each area being 60 m$^2$.

The application amounts of pure N, P, and potassium (K) for rice were 225, 112.5, and 255 kg ha$^{-1}$, respectively. For wheat season, the corresponding amounts were 180, 90, and 204 kg ha$^{-1}$, respectively. The N deficiencies in CS and CB were complemented by inorganic N fertilizer. The N, P, and K fertilizers used in the experiment were urea, calcium superphosphate, and potassium sulfate, respectively. Straw or biochar was added to the soil surface and thoroughly mixed with a depth of approximately 15 cm. Rice/wheat straw and the derived biochar were continuously sent back to the land as amendments in wheat/rice season. Pyrolysis was anaerobically performed at 500–600 °C in a vertical charcoal furnace (ECO-5000; Wuneng Environment Co., Ltd, Zhejiang, China) for biochar preparation. The basic characteristics of the straw and biochar are listed in Table S1.

## Soil sample collection and analysis

In June 2016, after wheat harvest, samples were aseptically taken from the 0–20 and 20–40 cm soil layers using a five-point sampling method. The samples were stored in polyethylene bags at low temperature and immediately returned to the laboratory. Then, the samples were sieved to remove roots, rocks, and litter. Approximately 100 g of sieved soil was subjected to different aggregate classifications and subsequently frozen at −80 °C for fungal community analysis. The residual portions of soil samples were further air-dried for physicochemical property determination.

Soil pH was measured by potentiometry with a soil-water ratio of 1:2.5 (Mettler Toledo FE20 Plus, Shanghai, China). Gravimetric moisture content was determined by weighing the soil, followed by drying the soil samples at 105 °C for 24 h and weighing the soil again. $NH_4^+$–N and $NO_3^-$–N were assayed with Nessler's reagent and the phenol disulfonic acid colorimetric methods, respectively. SOC and total nitrogen (TN) contents were determined with an elemental analyzer (Elementar, Langenselbold, Germany) and the KDN-08C automatic nitrogen determination instrument (Xinrui, Shanghai, China), respectively.

## Aggregate size distribution

Soil water-stable aggregates were separated by an agglomerate analyzer according to the wet sieving method with some modifications (Elliott, 1986). Briefly, a series of sieves was used to obtain three aggregate classes: macroaggregates (>0.25 mm), microaggregates (0.053–0.25 mm), and silt clay (<0.053 mm). First, the soil samples were slowly wetted for 5 min using sterile water. The analyzer was then shaken up and down at 20 times min$^{-1}$ for 5 min with the column submerged in water throughout the whole process. The fractions remaining on each sieve were collected at the end of sieving. Aggregates in the beakers were stored at −80 °C for microbial analysis, and the remaining portions were weighed to analyze the relative indices after being oven-dried.

The mean weight diameter (MWD) represents aggregate stability and was calculated using Eq. (1):

$$MWD = \sum_{i=1}^{n} X_i W_i \tag{1}$$

The geometric mean diameter (GMD) was calculated using Eq. (2):

$$\text{GMD} = \exp\left(\frac{\sum_{i=1}^{n} W_i \ln X_i}{\sum_{i=1}^{n} W_i}\right) \tag{2}$$

where $i$ corresponds to each collected fraction, and $X_i$ and $W_i$ refer to the mean diameter and the dry weight of $i$-size aggregates, respectively.

The fractal dimension ($D$) was obtained by regression analysis using Eq. (3):

$$(3-D)\lg\left(\frac{d_i}{d_{\max}}\right) = \lg\left[\frac{W_{(\delta \leq d_i)}}{W_0}\right] \tag{3}$$

where $d_i$, $d_{\max}$, $W_{(\delta \leq d_i)}$, and $W_0$ refer to the average diameter of an $i$-size aggregate, the maximum diameter of all aggregates tested, the weight of aggregates with particle size $< d_i$, and the total weight of aggregates, respectively. Finally, $D$ was obtained by regression analysis.

## Fungal DNA extraction and high-throughput pyrosequencing

Aggregate DNA was extracted using the MoBio PowerSoil Soil® DNA Isolation Kit (12888) according to the manufacturer's instructions. An aliquot of the DNA extracted from each sample was used as a template for amplification after its quality was confirmed by spectrophotometer (RS232G; Eppendorf, Hamburg, Germany). The internal transcribed spacer (ITS) regions were amplified with the primers ITS1F (5′-CTTGGTCA TTTAGAGGAAGTAA-3′) and ITS1R (5′-GCTGCGTTCTTCATCGATGC-3′). PCR amplification was conducted as follows: 30 s of initial denaturation at 98 °C, followed by 35 cycles of denaturation at 98 °C for 15 s, annealing at 50 °C for 30 s, extension at 72 °C for 30 s, and a final extension at 72 °C for 5 min. Sequencing was performed on a MiSeq PE300 platform at Majorbio Biotechnology Co. Ltd (Shanghai, China).

The original data fragments were quality-controlled using Trimmomatic software, merged with FLASH and analyzed using Quantitative Insights into Microbial Ecology software. Operational taxonomic units (OTUs) with 97% similarity were clustered using USEARCH 7.0 and analyzed against the UNITE database (Release 6.0 http://unite.ut.ee/index.php) with a confidence threshold of 70%. All sequences have been deposited in the NCBI Short Reads Archive database (Accession Number: SRP161464).

## Statistical analysis

The data were analyzed on the free online Majorbio I-Sanger Cloud Platform (www.i-sanger.com). Principal coordinates analysis (PCoA) was carried out to investigate the fungal community differences between four different treatments based on OTU results using R software. Significant differences between the trial treatments were determined by ANOVA analysis using SPSS 16.0 software (SPSS Corp., Chicago, IL, USA). Significant differences between means were determined with the least significant difference (LSD) test at the $p = 0.05$ level. Figures were prepared using ORIGIN 8.0 (OriginLab Corporation, Northampton, MA, USA).

**Table 1 Physicochemical characteristics of soil samples under different fertilization treatments and different depths.**

| Depth (cm) | Treatments | pH (H$_2$O) | SOC (g kg$^{-1}$) | TN (g kg$^{-1}$) | EC (mS cm$^{-1}$) |
|---|---|---|---|---|---|
| 0–20 | CK | 8.90 ± 0.16 a | 7.98 ± 1.51 b | 1.05 ± 0.19 | 0.05 ± 0.01 |
| | CF | 8.21 ± 0.55 ab | 9.09 ± 0.70 b | 0.91 ± 0.08 | 0.04 ± 0.02 |
| | CS | 8.01 ± 0.42 b | 9.61 ± 3.27 ab | 0.94 ± 0.07 | 0.05 ± 0.01 |
| | CB | 8.60 ± 0.24 ab | 13.08 ± 0.58 a | 0.97 ± 0.14 | 0.04 ± 0.00 |
| 20–40 | CK | 8.81 ± 0.27 | 3.31 ± 0.91 b′ | 0.86 ± 0.10 b′ | 0.04 ± 0.01 ab′ |
| | CF | 8.44 ± 0.53 | 3.10 ± 0.69 b′ | 0.85 ± 0.02 b′ | 0.05 ± 0.02 ab′ |
| | CS | 8.48 ± 0.25 | 5.01 ± 0.30 a′ | 1.13 ± 0.11 a′ | 0.03 ± 0.01 b′ |
| | CB | 8.73 ± 0.04 | 5.54 ± 1.17 a′ | 1.00 ± 0.06 a′ | 0.05 ± 0.01 a′ |

Note:
Data (average ± SE, $n = 3$) in the same column with different letters indicates significant differences according to LSD test ($p < 0.05$). The "a" and "a′" represent the ANOVA results for 0–20 and 20–40 cm soils, respectively.

# RESULTS

## Soil agrochemical properties

Soil nutrient contents were measured to investigate the influence of different fertilization treatments on soil physiochemical properties (Table 1). For the 0–20 cm soil depth, CB had the highest SOC content compared to CF and CK ($p < 0.05$); the difference with CS was not significant. Compared to CF, CS and CB both significantly increased the SOC and TN contents in 20–40 cm soil ($p < 0.05$). CS had a lower soil pH than CK in the 0–20 cm layer ($p < 0.05$), which may be due to more anions being released during straw decomposition. In 20–40 cm soil, CB had a higher electrical conductivity (EC) value than CS ($p < 0.05$).

## Water-stable aggregate distribution and stability analysis

Macroaggregates were the dominant aggregate size class in both 0–20 and 20–40 cm layers (Fig. 1). CS and CB, compared with CF, dramatically increased the macroaggregate content by 17.77–18.87% and 33.55–50.87% in the 0–20 and 20–40 cm soil layers, respectively, with CB performing more remarkably ($p < 0.05$). In terms of microaggregates, compared to CF, CS decreased its mass proportion in the 0–20 cm soil layer by 11.61%; CB reduced its content in 20–40 cm soil layer by 25.84% ($p < 0.05$). For silt clay content, compared to CF, CS and CB showed dramatic decreases of 39.46–66.12% and 40.40–50.99% in the surface soil and deep soil, respectively ($p < 0.05$).

Aggregate stability is an influential factor governing soil erodibility (*Ahmadi et al., 2011*). MWD and GMD have commonly been used as indicators to reflect agglomeration aggregation and the stability of soil aggregates. Compared to CF, CS and CB increased MWD and GMD values at soil depths of 0–20 cm (16.92–29.23%, 29.41–47.06%) and 20–40 cm (23.915–30.43%, 39.13–43.48%), respectively; CB exhibited a more prominent effect than CS ($p < 0.05$) (Table 2), indicating that organic amendment facilitates aggregate stability. D was used to indicate the size, distribution, and homogeneity of the soil particles; the higher the value, the poorer the soil permeability (*Ahmadi et al., 2011*). CS and CB remarkably reduced the *D*-value by 1.22–1.49% and 0.44–0.60% at the 0–20

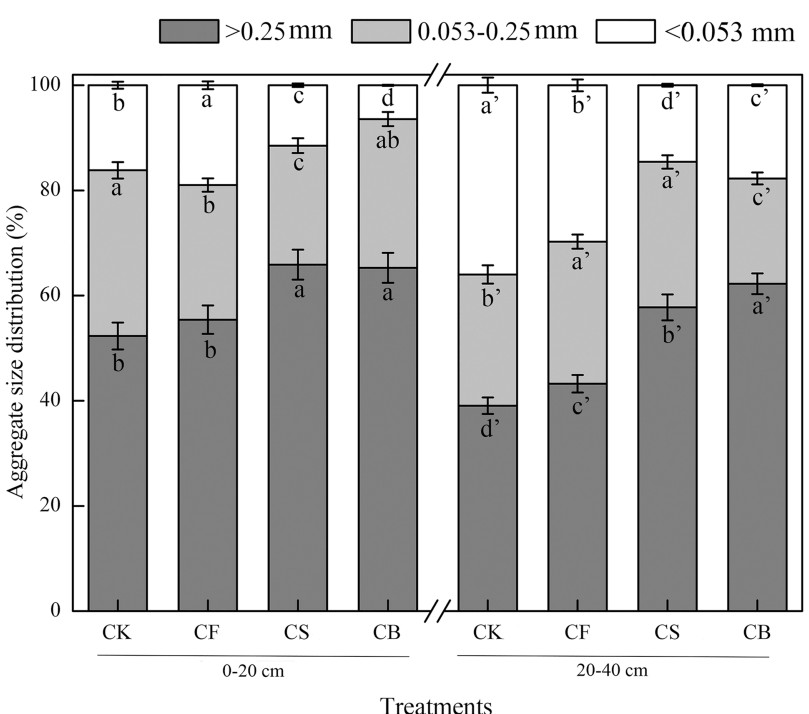

**Figure 1 Effects of different treatments on the distribution of soil water-stable aggregates.** Data here are mean ± SE, $n = 3$. Different lowercase letters in specific aggregate class refer to significant difference according to LSD test ($p < 0.05$). The "a" and "a'" represent the ANOVA results for 0–20 and 20–40 cm soils, respectively.

**Table 2 Effects of different treatments on the stability indices of soil aggregates.**

| Soil depth (cm) | Treatments | MWD (mm) | GMD (mm) | D |
|---|---|---|---|---|
| 0–20 | CK | 0.61 ± 0.03 c | 0.33 ± 0.01 c | 2.971 ± 0.054 a |
| | CF | 0.65 ± 0.03 c | 0.34 ± 0.02 c | 2.961 ± 0.072 a |
| | CS | 0.76 ± 0.04 b | 0.44 ± 0.03 b | 2.925 ± 0.036 b |
| | CB | 0.84 ± 0.04 a | 0.50 ± 0.03 a | 2.917 ± 0.074 b |
| 20–40 | CK | 0.38 ± 0.02 c' | 0.19 ± 0.01 c' | 2.987 ± 0.032 a' |
| | CF | 0.46 ± 0.03 b' | 0.23 ± 0.01 b' | 2.982 ± 0.048 a' |
| | CS | 0.57 ± 0.02 a' | 0.32 ± 0.02 a' | 2.969 ± 0.059 b' |
| | CB | 0.60 ± 0.02 a' | 0.33 ± 0.01 a' | 2.964 ± 0.035 b' |

**Note:**
Data in the table are mean ± SE, $n = 3$. Different lowercase letters in the same column refer to significant difference according to LSD test ($p < 0.05$). The "a" and "a'" represent the ANOVA results for 0–20 and 20–40 cm soils, respectively.

and 20–40 cm soil depths, respectively, which verified that straw and straw biochar promoted the uniform distribution of soil aggregates.

## Sequence data and alpha diversity

A total of 4,245,052 sequences with an average length of 258 bp were obtained from all 72 samples. Fertilization affected soil fungal abundance in different aggregates (Fig. S1). In the 0–20 cm soil layer, the number of fungal OTUs was inversely related to the aggregate size; CB harbored the smallest number of OTUs of the treatments (636, 831, and
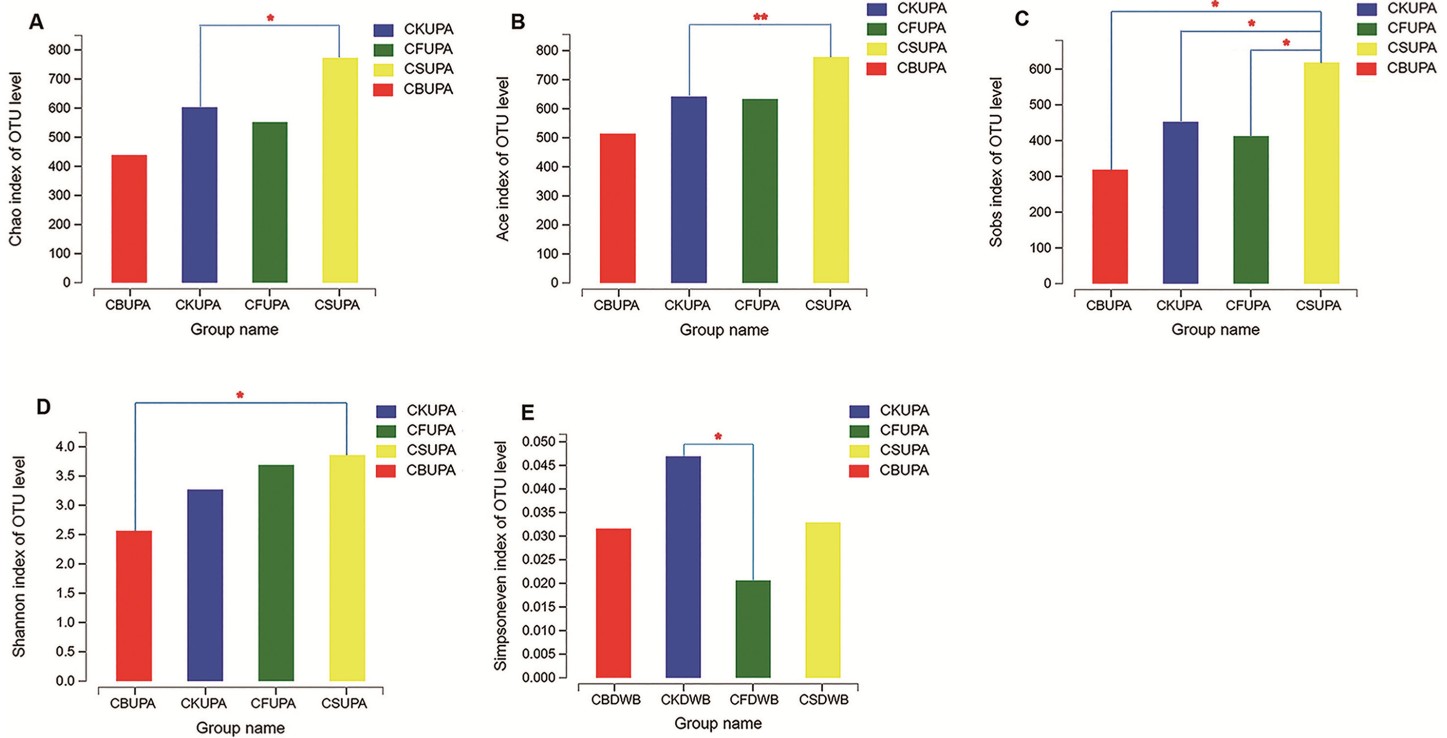

**Figure 2 Alpha diversity analysis between the four treatments (CK, CF, CS, CB) in 0–20 (UP) and 20–40 (DW) cm soils, respectively.** (A–D) The levels of Chao, Ace, Sobs, and Shannon of the macroaggregates in 0–20 cm soil layer, respectively; (E) exhibited the level of Simpsoneven of the microaggregates in 20–40 cm. The "A" and "B" in Group name refer to macroaggregate and microaggregate, respectively.

872 in macroaggregate, microaggregate, and silt clay, respectively), which was inconsistent with previous reports that the proportion of fungal groups in macroaggregates is high, indicating a major role in their formation. In the 20–40 cm soil, no such remarkable tendency was observed.

Alpha analysis of the fungal community under different treatments at different soil depths was estimated using Student's $t$-test (Fig. 2). The Chao1 estimator, abundance-based and coverage estimator (ACE), and observed richness (Sobs) were employed as community richness indices (*Liu et al., 2015*). Significant variances were observed among the four treatments for macroaggregates in surface soil. The Chao 1 and ACE values of CS were higher than those of CK ($p < 0.05$, $p < 0.01$, respectively); higher Sobs values were observed in CS than in CK, CF, and CB ($p < 0.05$). As for both microaggregates and silt clay, no significant difference was observed between the four treatments in the 0–20 cm soil layer, suggesting that fungi played a more important role in large aggregate formation in CS (*Li et al., 2018*). The Shannon and Simpsoneven indices, calculated by richness and species abundance, were used as diversity and evenness indices, respectively, within each individual sample. Compared to CS, CB decreased the Shannon index in macroaggregates in 0–20 cm soil ($p < 0.05$); no difference was observed with the other treatments. In deep-layer soil, microaggregates in CF had a dramatically lower Simpsoneven value than CK ($p < 0.05$), which

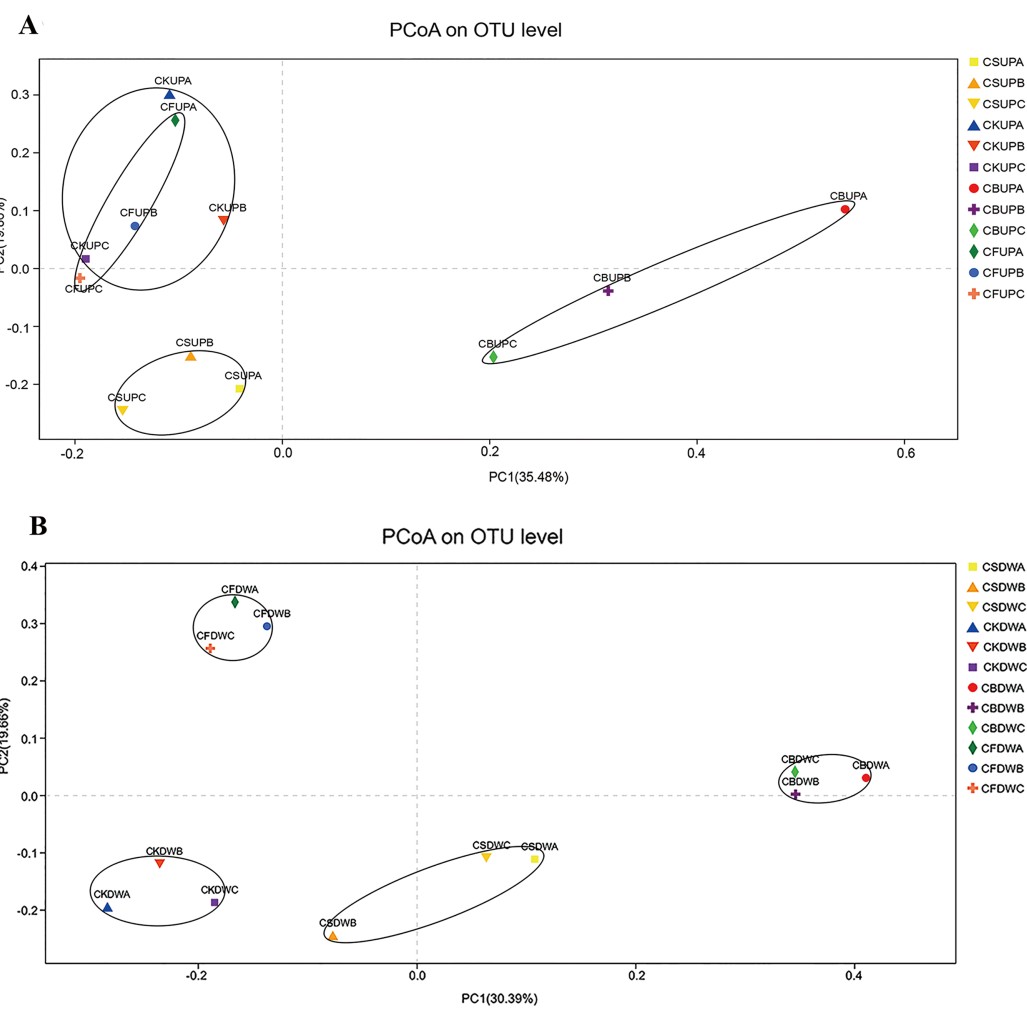

**Figure 3 Principal co-ordinates analysis (PCoA) of soil fungal community in four different fertilization treatments (CK, CF, CS, and CB) in 0–20 cm (A) and 20–40 cm (B) soil layers, respectively.** In the legend, "UP" and "DW" referred to the soil depths of 0–20 and 20–40 cm, respectively. "A," "B," and "C" meant macroaggregate, microaggregate, and silt clay, respectively.

suggested that long-term chemical fertilization probably impairs the evenness of the fungal community.

## Comparison of fungal community structure

PCoA was used to investigate the variations in fungal community structure caused by long-term fertilization and aggregate distribution (Fig. 3). For the surface soil, samples of CB were separated with the other three treatments by PC1 (35.48%); CS samples were gathered at the bottom left corner of the graph, apart from CK and CF, by PC2 (19.86%). CK and CF showed similar fungal community structures, as they were located in the upper part of PC2. With respect to the 20–40 cm soil layer, CF and CB were separated from CS and CK by PC2 (19.66%); PC1 (30.39%) approximately differentiated CB from CK and CF. Non-metric multidimensional scaling analysis based on Pearson distances,

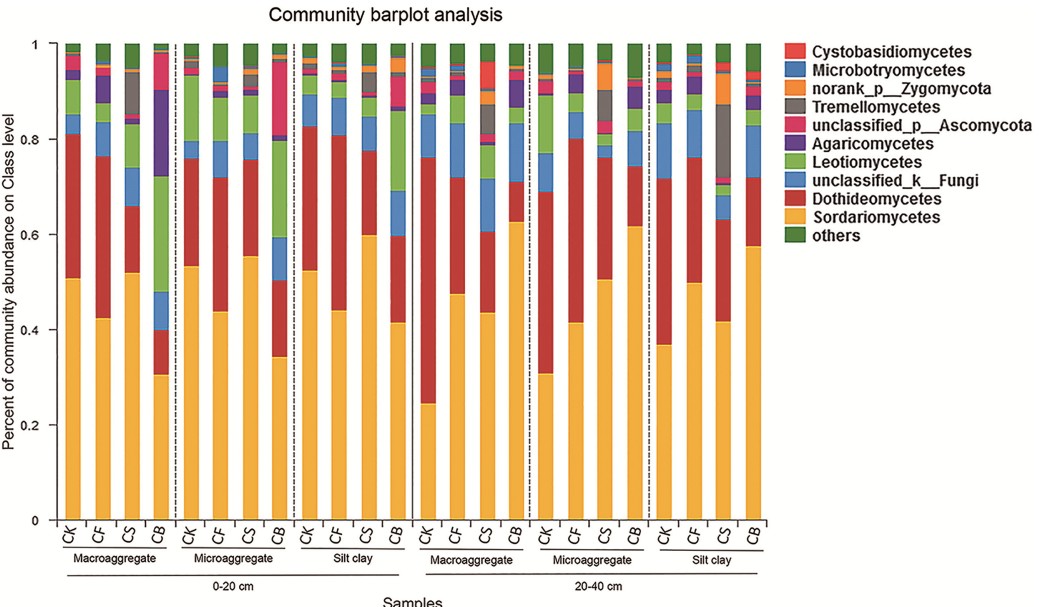

**Figure 4 Relative abundances of the dominant fungal classes for all the soil subsamples.** Relative abundances are based on the proportional frequencies of the DNA sequences that could be classified.

in accordance with the PCoA results, presented stress values of 0.098 and 0.091 for top soil (0–20 cm) and deep soil (20–40 cm), respectively (Fig. S2). The fungal community structures in CS and CB were significantly different from those in CF and CK at the 97% OTU level, and that of CS was significantly different from that of CB.

## Fungal community composition

Fertilization, agglomerate levels and sampling depth all can largely alter the community composition of soil fungi. In the top layer of soil, the phyla Ascomycota, unclassified_k_Fungi, and Basidiomycota constituted over 97% of sequences (Fig. S3). Ascomycota was the most prevalent phylum in the soil; its members are known for their ability to degrade lignin, but it also includes many sugar fungi that utilize simple substrates (*Chen et al., 2013*). The other less dominant phyla were Zygomycota, Glomeromycota, and Chytridiomycota. At the class level, fertilization management played an important role in the relative abundance of fungi (Figs. 4 and 5). Figures 5A–5J showed the statistically differences of the representative classes in the aggregates in 0–20 and 20–40 cm, respectively. In the 0–20 cm soil layer, CB, compared to CF, decreased Dothideomycetes in all three aggregates and Glomeromycetes in silt clay ($p < 0.05$). Leotiomycetes contents in macroaggregates and silt clay in the 0–20 cm soil were significantly higher in CB than CF ($p < 0.05$). A similar phenomenon was observed in the deep-layer (20–40 cm) soil to variable extents. Fertilization decreased and increased the relative abundances of Dothideomycetes (in both macroaggregates and silt clay) and Sordariomycetes (in macroaggregates), respectively, with CB significantly differing from CK ($p < 0.05$). In microaggregates, fertilizer application resulted in trends of decreased

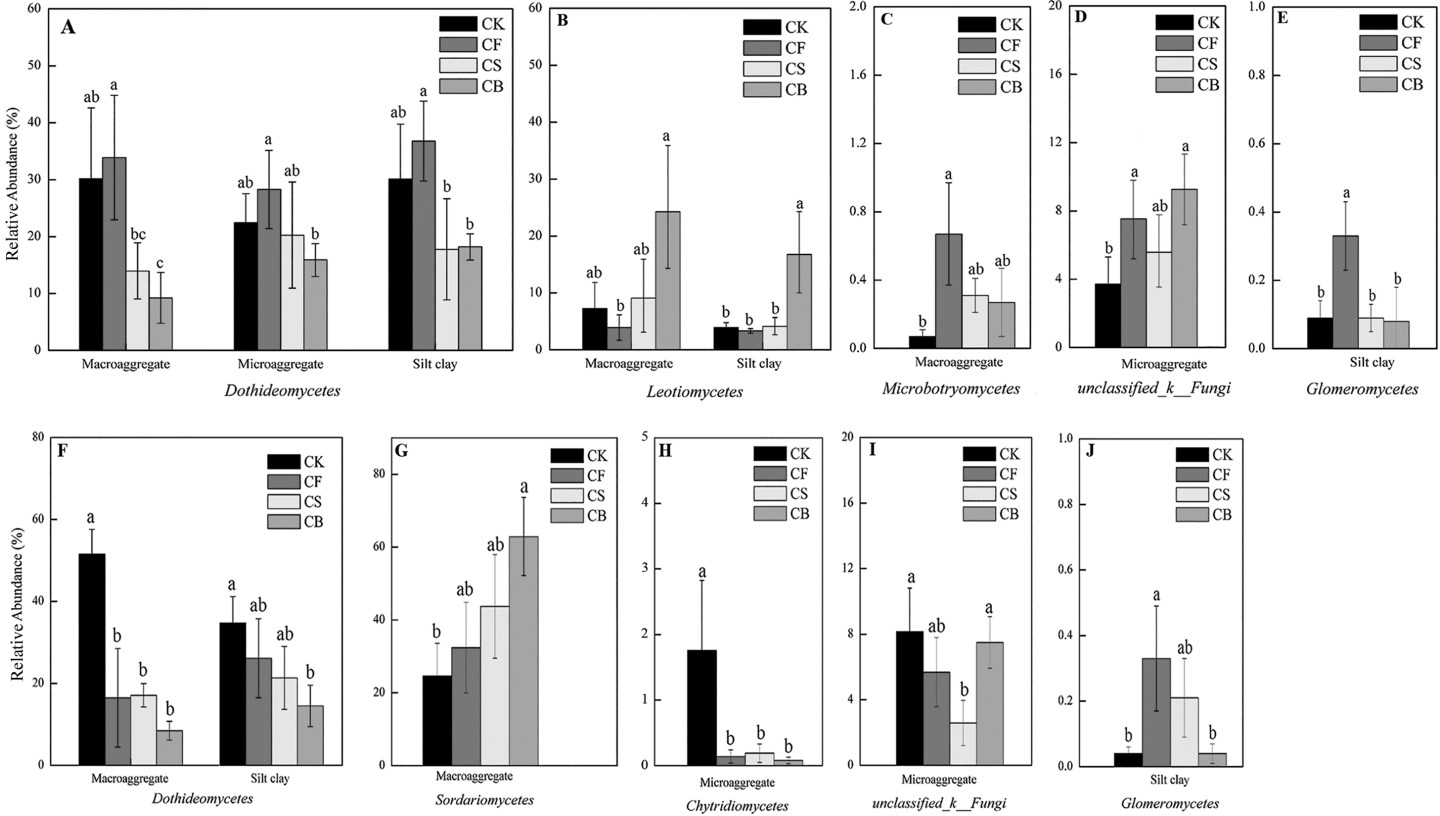

**Figure 5 Summarized statistically differences of the dominant classes in the soil aggregates in 0–20 (A–E) and 20–40 cm (F–J) soil.** Different lowercase letters at each aggregate distribution and each class indicate significant differences at $p < 0.05$ according to LSD test.

Chytridiomycetes and unclassified_k_Fungi. CF possessed higher Glomeromycetes content in silt clay than CK and CB ($p < 0.05$).

The statistical analysis results for the relevant abundances of the dominant fungal genera are shown in Table 3 (0–20 cm) and Table 4 (20–40 cm). At 0–20 cm, CB, compared to CF, reduced the contents of *Monographella* (macroaggregate), *Pyrenochaetopsis* (microaggregate and silt clay), and *Gibberella* (microaggregate) and increased *Articulospora* content (macroaggregate) ($p < 0.05$). CS also lowered the relative abundances of *Pyrenochaetopsis* (silt clay) and *Gibberella* (microaggregate) relative to CF ($p < 0.05$). As for the 20–40 cm soil, CB, compared with CF, decreased *Pyrenochaetopsis* (microaggregate and silt clay) and increased *Neurospora* (macroaggregate) ($p < 0.05$). CS had lower *Pyrenochaetopsis* (microaggregate) and higher *Neosetophoma* (silt clay) contents than CF ($p < 0.05$). Additionally, a small number of "unclassified" sequences were identified in this study, which means that these soils contained several unidentified fungi.

## DISCUSSION

Straw/biochar application tended to increase SOC and TN contents in both the 0–20 and 20–40 cm soil layers (Table 1). It seemed that CB treatment performed better than CS in our experiment. Biochar addition stimulates soil C sequestration by improving the

**Table 3 The summarized statistically relevant abundance differences of the dominant fungal genera in the surface soil (0–20 cm).**

| Items | Macroaggrergates | | | | Microaggregates | | | | Silt clay | | | |
|---|---|---|---|---|---|---|---|---|---|---|---|---|
| | CK | CF | CS | CB | CK | CF | CS | CB | CK | CF | CS | CB |
| *Pyrenochaetopsis* | a | ab | ab | b | ab | a | bc | c | ab | a | b | b |
| *Unclassified_c_Dothideomycetes* | | | | | | | | | a | b | b | b |
| *Unclassified_k_Fungi* | | | | | b | a | ab | a | | | | |
| *Unclassified_o_Pleosporales* | | | | | | | | | b | a | bc | c |
| *Gibberella* | | | | | ab | a | b | b | | | | |
| *Unclassified_o_Coniochaetales* | ab | ab | a | b | bc | a | ab | c | b | a | ab | b |
| *Monographella* | a | a | ab | b | | | | | | | | |
| *Articulospora* | ab | b | b | a | | | | | | | | |
| *Westerdykella* | ab | ab | a | b | | | | | | | | |

Note:
Different lowercase letters at each aggregate distribution and each fungal genus indicate significant differences at $p < 0.05$ according to LSD test.

**Table 4 The summarized statistically relevant abundance differences of the dominant fungal genera in the deep soil (20–40 cm).**

| | Macroaggrergates | | | | Microaggregates | | | | Silt clay | | | |
|---|---|---|---|---|---|---|---|---|---|---|---|---|
| | CK | CF | CS | CB | CK | CF | CS | CB | CK | CF | CS | CB |
| *Unclassified_k_Fungi* | | | | | a | ab | b | a | | | | |
| *Neurospora* | b | b | b | a | | | | | | | | |
| *Pyrenochaetopsis* | | | | | ab | a | b | b | ab | a | ab | b |
| *Neosetophoma* | | | | | | | | | a | b | a | b |

Note:
Different lowercase letters at each aggregate distribution and each fungal genus indicate significant differences at $p < 0.05$ according to LSD test.

aggregation, stabilization of soil organic matter with aggregates, and facilitating the physical protection of C (*Wang et al., 2017a*), which was more prominent than the results achieved with straw (*Huang et al., 2018*). Biochar addition can also reduce the amount of N leaching, due to its porous nature and large surface area, and hence increase soil nitrogen (*Xu et al., 2016*). Three fertilization treatments decreased the soil pH relative to CK to some extent (Table 1). However, the modulation of soil pH may be less pronounced in soils with a near neutral pH (*Chen et al., 2013*). The driving force behind a pH decrease is the oxidation of C to form acidic carboxyl groups, whereas an increase is likely related to the dissolution of alkaline minerals (*Lehmann et al., 2011*). Unlike laboratory incubations, field experiments are characterized by changing temperatures, frequency of tillage, and cycling of wetting and drying, which may decrease the impacts of biochar on both soil chemical and biological properties.

The results verified that extra organic biomass addition increased proportion of macroaggregates and reduced the portion of microaggregates and silt clay particles (Fig. 1). Biochar, produced at a lower temperature in our study, still contained a relatively large amount of nonpyrolyzed organic residue, which could potentially increase both

microbial activity and soil aggregation by providing feedstock for the production of extracellular polymeric substances acting as cementing agents for soil aggregates (*Wang et al., 2017a*). *Zhao et al. (2018)* reported similar results regarding the mass proportions of different aggregates with straw return to fields. However, their differences in the 20–40 cm layer were not significant, since the effects of fertilization on soil aggregation generally occur in the surface soil. Here, a significant difference was observed in 0–40 cm soil, which may be explained by years of returning straw to the field or soil conditions. *Wang et al. (2017a)* noticed that the addition of softwood biochar and walnut shell biochar both dramatically improved aggregate stability in Yolo soil (silty loam) but had no significant impact in Vina soil (fine sandy loam), which verified that the effects of biochar on soil quality are complicated.

Alpha diversity analysis showed that straw tended to increase fungal community richness and diversity, but straw biochar had the opposite tendency (Fig. 2). CF decreased fungal richness in this study, which was in contrast with the results reported by *Wang et al. (2017b)*. The C:N ratio in soil is among the strongest correlates of fungal community composition and related to community differentiation between fertilizer treatments (*Schlatter et al., 2017*). Organic nitrogen supplied via biochar may slow the fungal community response to increased surface residue and reduce the rate of residue breakdown (*Schlatter et al., 2017*). Small molecule compositions in organic matter can also strictly control the development and abundance of soil microbial communities, although they increase the soil total C and N (*Cozzolino et al., 2016*). Furthermore, the reduction in soil tensile strength by biochar addition may also make it physically easier for invertebrates to move through the soil, altering predator/prey dynamics and thus reducing fungal contents (*Atkinson, Fitzgerald & Hipps, 2010*; *Chen et al., 2013*), since bulk density has a positive correlation with bacterial biomass and a negative correlation with fungal biomass (*Jindo et al., 2012*). Thus, the effects of organic amendment are likely feedstock-, soil-, and plant-specific, although this conclusion requires further verification (*Lehmann et al., 2011*).

PCoA indicated that fertilization and soil aggregation both affected fungi community structure by Fig. 3. Several studies have reported similar tendencies. Both bacterial and fungal community composition were clearly separated by the straw addition and zero addition treatments (*Bei et al., 2018*; *Zheng et al., 2016*). *Ali et al. (2019)* also reported that soil microbiological communities differed from those of the control treatment group after biochar addition, meaning that biochar amendment altered the microbial community structure. However, the difference between the three hierarchical aggregates was not remarkable in the present study. Biochar had a more significant role in fungal community structure than CS (visualized by PCoA).

Fertilization methods have not been generally claimed to affect the microbial community (*Ding et al., 2017*; *He et al., 2008*; *Maarastawi et al., 2018*). *Wang et al. (2017b)* noted that CF could increase fungal richness; however, CF decreased fungal richness in the present study. Here, using the MiSeq sequencing platform, we analyzed the effects of different fertilization practices on fungal community composition and structure. At the

class level, Sordariomycetes and Dothideomycetes together occupied the dominant position for 39.98–82.68% (Fig. 4). Sordariomycetes was increased in macroaggregates in 20–40 cm soil (Fig. 5G), especially with the incorporation of biochar. Sordariomycetes was sensitive to fertilizers and responded to N addition with a nearly linear increase (*Mueller, Belnap & Kuske, 2015*). Fertilization seemed to decrease Dothideomycetes content in 0–40 cm soil, with the exception of CF in the 0–20 cm soil layer (Figs. 5A and 5F). Dothideomycetes, belonging to Ascomycota, is by far the largest and arguably most phylogenetically diverse class, containing a heterogeneous group of fungi that subsist in the majority of niches where fungi can be found. The best-known members of the group are plant pathogens that infect a broad range of hosts and cause severe crop losses (*Gnavi et al., 2014*; *Ohm et al., 2012*; *Schoch et al., 2009*). The relative abundance of Glomeromycetes, a fungal class regarded as widely symbiotic, decreased in response to drought (*Ochoa-Hueso et al., 2018*); thus, its content might have accordingly decreased during our sampling time (wheat season). Leotiomycetes, increased by CS and CB in macroaggregates and silt clay (0–20 cm soil), usually comprises plant-associated fungi whose ecological roles include those of pathogens, endophytes, symbionts, and saprobes, and a large number of taxa whose ecology and nutritional modes are poorly understood but are assumed to be plant-based (*Gnavi et al., 2014*; *Meerupati et al., 2013*).

*Pyrenochaetopsis* is one of the most abundant genera related to elevated atmosphere $CO_2$ levels (*Yu et al., 2018*). Although it was one of the most abundant genera in a rice paddy amended with biochar, the mass application of biochar would lower its relative abundance (*Zheng et al., 2016*). In our present study, straw and biochar continuously returned to the field for 6 years, rather than in a large mass, had similar effects on aggregates in both 0–20 and 20–40 cm soils, with the exception of macroaggregates in 20–40 cm layer (Table 3). Therefore, $CO_2$ emission variation in CS and CB needs further exploration. *Gibberella* and *Monographella*, which cause soil-borne diseases in crops (*Tatagiba, DaMatta & Rodrigues, 2016*; *Wang et al., 2018*; *Yan et al., 2018*), were lower in CS and CB in the 0–20 cm layer, with their levels in CB exhibiting significant differences relative to those in CF ($p < 0.05$). These reduced pathogenic fungal contents might lessen the probability of morbidity. From this perspective, the application of organic amendments is a win–win strategy for the management of diseases caused by soil-borne pathogens (*Bonanomi et al., 2010*). Biochar amendment increased the content of *Articulospora*, which was dominant in decomposing plant litter (*Seena et al., 2012*), in macroaggregates (0–20 cm). *Neurospora* is known particularly for its abundant presence in ecosystems following fire (*Schlatter et al., 2017*); it was greatly enriched by straw biochar to decompose burnt biosolids in the present study (Table 4). Under our experimental conditions, these remarkable variations in fungal genera are likely related to and might be indicators of the effects of biochar amendment. CS also exhibited reactions corresponding to these genera to different extents.

In conclusion, straw and straw-derived biochar applied with inorganic fertilizer affected fungal community structure and may have overcome the adverse effects of inorganic fertilizer.

## CONCLUSIONS

We conclude that the 6-year amendment of straw and straw-derived biochar improved soil physicochemical properties and altered fungal community structure in a rice–wheat rotation system to different extents. With respect to aggregation and compared with CF, CS and CB increased the content of macroaggregates (>0.25 mm) and decreased the contents of microaggregates (0.053–0.25 mm) and silt clay (<0.053 mm), with CB performing more prominently than CS ($p < 0.05$). In addition, CS increased fungal community richness and diversity, but CB decreased them. At the genus level, CB significantly increased fungi that decompose biosolids (*Articulospora* and *Neurospora*) and decreased pathogen fungi (*Monographella* and *Gibberella*) and $CO_2$-emission-related fungi (*Pyrenochaetopsis*) in specific aggregates ($p < 0.05$). These remarkable variations in fungal genera are likely related to and might be indicators of the effects of biochar amendment. Furthermore, it seemed that fertilization management played an important role in fungal community structure and aggregation.

## ACKNOWLEDGEMENTS

The authors are grateful for the anonymous reviewers for their constructive comments in improving both the language and scientific quality of the manuscript.

### Funding

This work was supported by the National Natural Science Foundation of China [41501259]; Shanghai Sailing Program [18YF1420900]; the SAAS Program for Excellent Research Team [nong ke chuang 2017 (A-03)]; Shanghai Agriculture Applied Technology Development Program [T20170105]; the Key Agricultural Technology Program of Shanghai Science and Technology Commission [16391901500]; Domestic Cooperation program of Shanghai Science and Technology Commission [18295810500]. The funders had no role in study design, data collection and analysis, decision to publish, or preparation of the manuscript.

### Grant Disclosures

The following grant information was disclosed by the authors:
National Natural Science Foundation of China: 41501259.
Shanghai Sailing Program: 18YF1420900.
SAAS Program for Excellent Research Team: nong ke chuang 2017 (A-03).
Shanghai Agriculture Applied Technology Development Program: T20170105.
Key Agricultural Technology Program of Shanghai Science and Technology Commission: 16391901500.
Domestic Cooperation program of Shanghai Science and Technology Commission: 18295810500.

## Competing Interests

The authors declare that they have no competing interests.

## Author Contributions

- Naling Bai conceived and designed the experiments, prepared figures and/or tables, authored or reviewed drafts of the paper, approved the final draft.
- Hanlin Zhang conceived and designed the experiments, prepared figures and/or tables, authored or reviewed drafts of the paper, approved the final draft.
- Shuangxi Li analyzed the data, contributed reagents/materials/analysis tools.
- Xianqing Zheng analyzed the data, contributed reagents/materials/analysis tools.
- Juanqin Zhang analyzed the data, contributed reagents/materials/analysis tools.
- Haiyun Zhang analyzed the data, contributed reagents/materials/analysis tools.
- Sheng Zhou performed the experiments.
- Huifeng Sun performed the experiments.
- Weiguang Lv conceived and designed the experiments, approved the final draft.

## DNA Deposition

The following information was supplied regarding the deposition of DNA sequences:

All sequences are available at the NCBI Short Reads Archive database (Accession Number: SRP161464).

## Data Availability

Bai, Naling; Zhang, Hanlin; LV, Weiguang; Li, Shuangxi; Zhang, Juanqin; Zhang, Haiyun; et al. (2018): Long-term effects of straw and straw-derived biochar on soil aggregation and fungal community in a rice–wheat rotation system. figshare. Fileset. https://doi.org/10.6084/m9.figshare.7205393.v1

## Supplemental Information

Supplemental information for this article can be found online at http://dx.doi.org/10.7717/peerj.6171#supplemental-information.

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
