# Peer review of "Long-term effects of straw and straw-derived biochar on soil aggregation and fungal community in a rice–wheat rotation system"

_PeerJ, doi:10.7717/peerj.6171_

## Round 0.1 · original submission · Major Revisions

As you can see below, both reviewers have made suggestions how your manuscript could be further improved. Please consider changing your manuscript according to the reviewers' suggestions, so it can be further considered for publication.

Reviewer 1 ·

Basic reporting

This manuscript is generally well-written using clear and professional English. The manuscript can be further improved regarding the following aspects: 1) define abbreviations when the term is firstly used, such as C (line 67) and TN (line 138). These terms might be commonly used in this area but it would be better to get consistent as the authors did a good job defining other abbreviations throughout the manuscript; 2) be explicit when describing methods or results. For example, in the abstract, the author said "soil samples of different aggregate fractions...". It should be explained what are different fractions. Is it based on sizes? Another example is "...CO2-emission-related fungi in specific aggregates (p<0.05)." What are the specific aggregates. 3) a few confusing statements containing misconceptions. In Line 64-66, it reads like bacteria are a branch of fungi, which is not the case.

Experimental design

The reviewer has a major concern about the experimental design because: 1) the author did not clearly describe how many types of biochars were applied in this study and how frequently applied. The reviewer guesses there should be two biochars according to the supplemental table, and the biochars were yearly buried instead of burying them only during the first or first two years as the crop biomass wastes were yearly generated so they will be yearly sent back to land as amendments. If so, the study would be interesting but becoming complicated as there are two types of biochars making influences at the same time without knowing which one had more impacts than the other. Also, as time goes on, the biochar/straw application rate, on an area basis, would increase since the carbon stayed on the land and accumulated year after year. These factors should be considered and discussed in this manuscript.

2) The manuscript lacks some important soil property descriptions as baseline or background data. For example, the soil texture plays an important role in soil aggregation. If the soil is clayey, many studies showed that soil amendment could not significantly increase aggregation. Is the soil used in this study sandy or silty? It would be interesting to know. Also, it would be better to have some data about the soil microbial data in year one or before the crop wastes were applied. Basically, the authors were comparing CS and CB with CF and CK during the sixth year and concluding what kind of impacts biochar or straw had on the microbial parameters. Without knowing the initial baseline data, the conclusions can be misleading.

Validity of the findings

The results and discussion are generally well-written and closely correlated but in some statements, it would be better to point out which figure or table supports such sentences. For example, line 266-268, "C:N ratio was among the strongest correlates of fungal community..." Which figure could prove this?

·

Basic reporting

The research topic is very important. Understanding the long-term impacts of biochar at field scale is critical for determining whether biochar can be used as an soil management tool in the future. At the same time, there is very few research looked into the impact of management practice on microbial community in soil water-stable aggregate fractions.
The manuscript is written in fine English. The related and latest references were sufficiently and properly cited in this manuscript.
The results of this study is presented clearly and well discussed.

Experimental design

The research question is clear and well defined. It's helpful to fill the current knowledge gap in biochar field application.
Methods are well described and sufficient details of methods were provided.

Validity of the findings

The experiment was properly conducted including control and replicates. Data provided is robust and can support discussion and conclusion in this manuscript.

Additional comments

Here a couple of specific comments:
Specific comments:
Line 64-66. Seems like some part of this sentence is missing or half modified. Please check.
Line 66. Provide nutrients for?
Line 87. The impact of biochar on soil properties and microbial activities were not always positive. Please modify this paragraph and include some literature with negative impacts, too.
Line 103. Please also provide some hypotheses here.
Line 116. Please state what pure nitrogen meant here. Was it total N or plant available N?
Line 138. Was soil inorganic C tested here?
Figure 1. This figure looks redundancy to me since the mean weight diameter of soil aggregates were reported already.

---

## Round 0.2 · accepted · Accept

Thank you for carefully revising your manuscript and addressing the reviewer comments. I look forward to publication of your paper and wish you all the best in your future research!

#